

# The effect of three different pre-match warm-up structures on male professional soccer players' physical fitness

Mehdi Ben Brahim[1], Alejandro Sal-de-Rellán[2], Adrián García-Valverde[2], Hussain Yasin[1] and Javier Raya-González[3]

[1] Health and Physical Education Department, Prince Sultan University, Riyadh, Kingdom of Saudi Arabia
[2] Faculty of Health Sciences, Universidad Internacional Isabel I de Castilla, Burgos, Spain
[3] Faculty of Sport Sciences, University of Extremadura, Cáceres, Spain

## ABSTRACT

**Background.** Optimizing the soccer players' match preparation is one of the most relevant concerns of coaches for strength and conditioning training. Thus, the aim of this study was to analyze the effects of three pre-match warm-up structures on the physical condition of male professional soccer players.

**Materials & Methods.** Seventeen male professional soccer players (age: 20.9 ± 2.4 years) of one Tunisian Professional Soccer League team participated in this study. Players performed three times a typical pre-match warm-up (WU) [aerobic phase (AE); neuromuscular phase (NM); velocity phase (VL); and ball-specific phase (BS) variating the order of the included phases/exercises: WU1 (AE + BS + NM + VL); WU2 (AE + NM + VL + BS); and WU3 (AE + VL + BS + NM). After each warm-up phase, players completed the following physical fitness tests: linear sprint test, countermovement jump test, 15-m ball dribbling agility test and ball shooting test. Additionally, the rate of perceived exertion (RPE) was registered after each warm-up.

**Results.** The results indicated significant differences between WU1 and WU3, with better results in the ball shooting test and RPE in WU3. In addition, there were no significant differences in the other variables and between other warm-up structures. These findings could be of great interest for coaches to the strength and conditioning coaches for professional soccer teams in order to improve the players physical performance and perceived exertion.

## INTRODUCTION

Soccer is considered as an intermittent team-sport (*Altmann et al., 2020*), characterized by the combination of a large number of high-intensity actions (*e.g.*, accelerations, decelerations, changing of direction, jumping or kicking the ball) with low-intensity actions (*Krustrup & Bangsbo, 2001*; *Nobari et al., 2023*). Per match, soccer players typically cover a total distance of 10–13 km (*Modric et al., 2022*), in which players might cover a distance of 190–185 m at sprint and reach a maximum velocity of 31 km h$^{-1}$. Additionally, soccer players cover near to 418–568 m per match at high-speed running (*i.e.*, 21–24 km h$^{-1}$)

Corresponding author
Alejandro Sal-de-Rellán,
salderellanguerra@gmail.com

(*Nobari et al., 2022*; *Rey et al., 2023*) and perform around 100–150 accelerations (*Varley & Aughey, 2013*). These results support the notion that soccer is a highly demanding sport (*Modric et al., 2022*), with variations in demands based on playing position or age (*Sal de Rellán-Guerra et al., 2019*). Given this, the pre-match warm-up is considered a key strategy in soccer programs, so optimizing it might be essential for soccer's player (*Olivares-Jabalera et al., 2021*). Warm-up is intended to help achieve optimal performance (*McGowan et al., 2015*) as well as a reduced risk of injury due to increasing muscle temperature, metabolic reactions, blood flow, nerve conductivity, anaerobic energy provision, oxygen consumption and force expression (*Bishop, 2003a*; *Bishop, 2003b*). Furthermore, several studies (*Bishop, 2003b*; *Bishop, 2003a*; *McGowan et al., 2015*; *Yanci et al., 2019*) have concluded that warm-up not only prepares the player physically but also have positive psychological effects such as self-confidence and concentration. Therefore, the aim of the warm-up would be twofold: (i) to prepare the players physically and mentally for the competition (*Bishop, 2003a*; *Fradkin, Zazryn & Smoliga, 2010*) and (ii) to prevent injuries (*Bishop, 2003b*).

Currently, many professional soccer teams follow warm up guidelines provides by programs such as FIFA 11 + prevention programme (*Nuhu et al., 2021*) or the Sportsmetrics Warm-up for Injury Prevention and Performance (*Grandstrand et al., 2006*). These warm-up protocols typically include running, static and dynamic stretching (*Fletcher & Monte-Colombo, 2010*), neuromuscular activities to prevent injury, high-intensity performance and post-activation exercises, and exercises with specific tactical objectives (*Bizzini et al., 2013*; *Chiu et al., 2022*; *Nuhu et al., 2021*). The aim of these warm-ups is to progress from less specific to more game-specific tasks to enhance performance (*Thapa et al., 2023*). On the other hand, some authors have suggested that a warm-up should begin with a general cardiovascular phase combined with dynamic stretching, and finish with sport-specific actions (*Sander et al., 2013*; *Taylor, Weston & Portas, 2013*) in order to stimulate the main systems and organs involved during the training session or during matches, through the reproduction of sporting gestures similar to competition (*Bishop, 2003b*; *Wakeling, 2004*). However, several variables related to warm-up configuration must be considered. Regarding warm-up duration, some authors found that both short warm-up and long warm-up were equally effective on repeated sprint performance (*Van den Tillaar, Lerberg & Von Heimburg, 2019*; *Van den Tillaar & Von Heimburg, 2016*). Specifically, *Taylor, Weston & Portas (2013)* concluded that it is more practical to perform a short warm-up which will allow the remaining time to be used for tactical preparation before the match. Other variables such as duration (*Yanci et al., 2019*), intensity (*Patti et al., 2022*; *Zois et al., 2011*; *Zois, Bishop & Aughey, 2015*) or rest protocols between warm-up and performance (*Dawson et al., 1997*) have been analyzed, obtaining that the short warm-up (*i.e.,* 8 min) is the only one that improves acceleration without increase related to RPE. However, none of the aforementioned studies have specifically analyzed pre-match warm-up. In this regard, *Bishop (2003a)* argued that the content, intensity and duration of warm-up should vary according to the physical and physiological demands of the competition, so the study of the effects of different warm-up structures would be guaranteed.

Given the lack of consensus regarding the optimal configuration of pre-match warm-up strategies during soccer matches (*Dawson et al., 1997*; *Patti et al., 2022*; *Yanci et al., 2019*; *Zois et al., 2011*; *Zois, Bishop & Aughey, 2015*), the purpose of this study was to analyze the effects of three pre-match warm-up structures on the physical condition of male professional soccer players. Based on articles mentioned above, we hypothesize that the order of the exercises during the warm-up can have significant differences in RPE and match performance.

## MATERIALS & METHODS

### Participants

Seventeen male professional soccer players (age: 20.9 ± 2.4 years; height: 170 ± 0.5 cm; body mass: 70.8 ± 20.4 kg; systematic practice: 6 ± 2 years) voluntarily participated in this study. Soccer players belonged to the same team of the Tunisian Professional League during the experimental period. An a priori power analysis (G*Power, v3.1.9.2; Universität Kiel, Kiel, Germany) indicated a sample size of at least 15 was required to achieve power (1- $\beta$) of 0.80 with an effect size (ES) of 0.30 (moderate effect) and alpha of 0.05. Players who had suffered injuries within the previous 3 months or had recently joined with the club the club within the last six months were excluded to the subsequent analysis. Prior to star the experimental phase, participants were provided with information regarding the study's aims and potential risks, and they provided written informed consent. Futhermore, the study was conducted in accordance with the principles outlined in the Declaration of Helsinki (2013) and approved by the ethics committee of the Isabel I de Castilla university (FUi1-PI002).

### Measures

A crossover randomized design was applied to analyze the effects of three pre-match warm-up structures on male professional soccer players' physical fitness. Professional soccer players completed three assessment sessions after a structured warm-up, which were applied in different weeks during a training camp. In each experimental session, warm-ups were completed following a crossover randomized order to avoid any effects of factor time or session on the results. Additionally, rate of perceived exertion (RPE) was registered after warm-ups. Players were familiarized with warm-up exercises and the fitness test battery since all of them were part of their daily soccer routines. To avoid interference effects, all assessment sessions were performed at the same time day (*i.e.,* 9:30 a.m.) and under similar environmental conditions (23–25 °C), with the same sports clothes, and by the same testers. Also, all warm-ups were performed in the last day of each microcycle (*i.e.,* Sunday). Players were encouraged to maintain their nutritional routines avoiding caffeine-rich drinks such as coffee prior to assessments.

### Design and procedures
### Warm-up protocol

To compose the warm-up protocol (24 min), the following phases were included (File S1): (i) aerobic phase (AE; 10 min), comprising jogging (4 min), running and mobility
exercises (*e.g.*, skipping, 4 min) as well as 2 min of dynamic stretching focused on lower limbs; (ii) neuromuscular phase (NM; 4 min), including strengthening (*e.g.*, half-squat, 2 min) and balance exercises (*e.g.*, unilateral receptions, 2 min); (iii) velocity phase (VL; 2 min), comprising linear (two sprints with 30 s of rest) and non-linear sprint (two sprints with 30 s of rest) over several distances (5 and 10 m); and (iiii) ball-specific phase (BS; 8 min), including passes in several distances (from 5 to 35 m, 4 min) and a 5vs5 small sided game (20 × 20 m, 2 sets of 1 min 30 s with 1 min of rest) . For this study, three different structures based on this warm-up protocol were proposed: WU1 (AE + BS + NM + VL); WU2 (AE + NM + VL + BS); and WU3 (AE + VL + BS + NM).

## Battery fitness test

*Linear sprint test.* Participants were encouraged to complete two maximum sprints of 20 m, with a split at 5 m being allowed a 2 min of passive recovery between attempts. The players started from a standing position, 0.5 m behind the first set of photoelectric cells (Polifemo Light Radio, Microgate, Bolzano, Italy), before running at maximal speed to the second photoelectric cell. The fastest sprint was recorded and the maximum speed was calculated.

*Countermovement jump test.* Players completed three bilateral Countermovement Jumps (CMJ) with a 45-second passive recovery period between each jump. During the CMJ, players were instructed to perform a downward movement followed by a complete, explosive extension of the lower limbs, while keeping their arms akimbo. Each jump was recorded at a sampling rate of 240 Hz using an iPhone 8 Plus mobile device (Apple Inc, Cupertino, CA, USA) and the My Jump 2.0 mobile application was used to measure jump height. This application has been previously validated ( $r = 0.99$ ) and demonstrated good reliability (intraclass correlation coefficient = 0.99) for assessing CMJ height (*Balsalobre-Fernández, Glaister & Lockey, 2015*). The highest jump recorded was selected for subsequent analysis.

*15-m ball dribbling agility test.* This test was performed according to the protocol previously described by *Mujika et al. (2009)*. For register the time employed to cover the test, photoelectric cells (Polifemo Light Radio; Microgate, Bolzano, Italy) were used. Two maximal repetitions with 3 min of passive recovery were allowed for each player. Players were required to drive a ball while performing the test. After the slalom section, the ball was kicked under the hurdle while the player cleared it. The player then freely kicked the ball towards either of two small goals placed diagonally 7 m on the left and the right sides of the hurdle, and sprinted to the finish line. The fastest repetition was selected for the further analysis.

*Ball shooting test.* To assess ball velocity, players performed five ball shoots with a 1-minute rest interval between each attempt. They were instructed to strike the ball as hard as possible using an instep kick technique from a stopped ball position (*Peráček et al., 2018*). The target for the ball shoot was a 1 × 1 meter area, and players aimed to achieve the maximum velocity of the ball towards the target. Ball speed was measured by a radar gun located 3 m

from the stopped ball and pointed toward the target according to the instruction manual (Sports Radar Gun SRA 3000; Precision Training Instrument, Arlington Heights, IL, USA). Participants in the study wore their own soccer shoes, and a standard ball was used for the ball shoots (Adidas, Germany; $69.0 \pm 0.2$ cm in circumference and $440 \pm 0.2$ g in mass). The highest recorded ball speed among the five attempt was selected for further analysis.

## Subjective load measure

*Rate of perceived exertion.* After each warm-up, players were asked to provide their perceived exertion of each modality. Players answered the question, "How hard was your warm-up?" 10 min after every warm-up, and the data were collected by the same person (*i.e.*, physical trainer). The intensity was determined using the Foster 0–10 scale (*Raya-González et al., 2020*). Each player was confidentially interviewed and could not see the values rated by the other participants. All players were previously familiarized with this method as part of their training routine.

## Statistical analysis

Descriptive data are presented as mean $\pm$ standard deviations (SD). To assess the normality of data distribution and the homogeneity of variances, the Kolmogorov–Smirnov and Levene tests were conducted, respectively. Since all analyzed variables had a normal distribution, parametric tests were applied. A repeated measure ANOVA was applied to detect differences in acute performance after the application of warm-ups and the post hoc's with Bonferroni corrections was applied in those variables which showed significant differences. The Hedges'g was used to calculate the magnitude of differences, which was interpreted as trivial ($g < 0.2$), small ($g < 0.5$), moderate ($g < 0.8$) and large ($g \geq 0.8$). Data was analyzed using the Statistical Package for Social Sciences (SPSS 25.0, SPSS Inc., Chicago, IL, USA), and the statistical significance was set at $p < 0.05$.

# RESULTS

In Table 1 differences in physical test and RPE regarding the applied warm-up are presented. Significant differences ($p < 0.05$) in ball velocity and RPE between WU1 and WU3 were observed. However, no significant differences were reported in the other variables and warm-up modalities.

# DISCUSSION

The aim of this study was to analyze the effects of three pre-match warm-up structures on the physical condition of male professional soccer players. This research holds significance for strength and conditioning coaches since it might provide valuable information on which pre-match warm-up structure is the most suitable for enhancing soccer players' match performance. Moreover, it has not yet been analyzed in professional soccer players. Remarkable results were found on WU1 and WU3 comparison, with better results in Ball-speed and RPE in WU3. In addition, there were no significant differences in the other variables and between other warm-up structures.

**Table 1 Descriptive data and differences between warm-ups regarding physical tests and rate of perceived exertion.**

| Variables | Mean ± SD | | | Pairwise comparisons (p; ES) | | |
|---|---|---|---|---|---|---|
| | WU1 | WU2 | WU3 | WU1 vs WU2 | WU1 vs WU3 | WU2 vs WU3 |
| SP5 (s) | 1.07 ± 0.06 | 1.09 ± 0.09 | 1.08 ± 0.08 | 1.00; 0.22 | 1.00; 0.13 | 1.00; 0.11 |
| SP20 (s) | 3.46 ± 0.24 | 3.48 ± 0.30 | 3.49 ± 0.35 | 1.00; 0.07 | 1.00; 0.09 | 1.00; 0.03 |
| CMJ (cm) | 31.11 ± 3.8 | 31.84 ± 3.61 | 32.51 ± 4.00 | 1.00; 0.20 | 0.21; 0.35 | 1.00; 0.17 |
| Ag (s) | 4.77 ± 0.35 | 4.67 ± 0.35 | 4.71 ± 0.45 | 1.00; 0.29 | 1.00; 0.17 | 1.00; 0.09 |
| Ball vel (km/h) | 76.08 ± 4.28 | 79.87 ± 5.67 | 82.53 ± 4.87 | 0.18; 0.67 | 0.01[*]; 1.32 | 0.54; 0.55 |
| RPE | 3.5 ± 0.6 | 3.1 ± 0.9 | 2.7 ± 0.8 | 0.45; 0.67 | 0.02[*]; 1.33 | 0.45; 0.44 |

Notes.

Abbreviations: SD, standard deviation; ES, effect size; SP5, time to cover a 5 m distance; SP20, time to cover a 20 m distance; CMJ, countermovement jump; Ag, 15-m ball dribbling agility test; Ball vel, ball shooting test; RPE, rate of perceived exertion.

[*]Significant differences ($p < 0.05$).

Our study revealed no significant difference in the time to cover a 5 m distance (SP5) and time to cover a 20 m distance (SP20) regarding any of the warm-up structures. Despite this, the obtained findings provide valuable information, since any strategy that increases sprinting ability in elite soccer is of great interest to the strength and conditioning coaches, due to the great influence of this variable on soccer performance, being the most frequent action that precede a goal (*Faude et al., 2017*). Regarding CMJ, non-significant differences were observed between the three types of warm-ups. Similarly, no significant differences in the 15-m ball dribbling agility test (Ag) were observed between the three warm-up structures. *Sheppard & Young (2006)* highlighted that this ability is influenced by balance, strength level, flexibility and muscular coordination. In addition, in this study agility was measured with the Ag, which in turn has a close relationship with technical skills such as dribbling and the sprint-15m (*Mujika et al., 2009*), which could influence the obtained results.

Our findings revealed significant differences in ball velocity between WU1 and WU3, indicating that WU3 improved the player's performance. This result can be attributed to the post-activation performance enhancement phenomenon, where an increment in voluntary strength production has been observed shortly after high muscle activation demands (*Blazevich & Babault, 2019*). Additionally, *Ali et al. (2007)* concluded that shooting performance remains consistent throughout the 90 min of the match, suggesting that a warm-up that activates this skill from the beginning could significantly impact the overall result. Coaches must consider this finding, since fast ball striking leads to a shorter reaction time of the opposing goalkeeper, ultimately increasing the likelihood of scoring goals (*Anzer, Bauer & Brefeld, 2021*). Therefore, it would be relevant for the success in professional soccer, as the primary objective in this sport is to score more goals than the opponent (*Russell, Benton & Kingsley, 2010*).

In terms of RPE, significant differences were observed between WU1 and WU3, with a lower RPE in WU3. This finding holds great relevance since RPE reflects the subjective perception of professional soccer players regarding to a specific task. Previous studies (*Van den Tillaar, Lerberg & Von Heimburg, 2019*; *Van den Tillaar & Von Heimburg, 2016*; *Yanci et al., 2019*) concluded that RPE tends to increase with the duration of the warm-up;

however, these studies did not consider the order of exercises. Upon reviewing these studies, it becomes apparent that all proposed warm-ups concluded with a speed phase, similar to WU1, which may result in a higher RPE compared to WU2 and WU3. Our results suggest that WU3, which concludes with the neuromuscular phase (NM), leads to improved performance in SP5 and SP20 due to the lower RPE reported by professional soccer players. In contrast, *Chiu et al. (2022)* compared a typical dynamic warm-up (*i.e.,* running with various tempos and dynamic stretching) with the FIFA 11 + warm-up (*i.e.,* running exercises with dynamic stretching; strength, plyometric and balance exercises; and high-intensity running exercises), and found no significant differences in RPE between the two protocols. Conversely, our results indicated that finishing with NM phase results in a lower RPE compared to finish with VL or BS phases. In this sense, a cognitive enough demand could have an influence on a player's fatigue perception (*Smith et al., 2019*). In fact, several authors have shown that mental tasks are able to affect physical performance (*Bray et al., 2012*) and also a physiological response (*Greig et al., 2007*). Therefore, the warm-up tasks need to be cautiously selected to improve human performance ability perception.

This study presents some limitations that should be known by practitioners. The main limitation is that carrying on the study in different weeks during the in-season period the physical condition of players might vary between the three different measurements. Also, the collected data was from one team, so the sample is small. Another limitation is that other variables such as quality of sleep or muscle pain, which could influence the RPE, were not analyzed. Finally, the position of each player on the field was not considered. This could lead to variations in the warm-up objectives because the physical demands in each playing position are different (*Modric et al., 2022*).

## CONCLUSIONS

In conclusion, it seems that a warm-up protocol following the order of WU3 (aerobic phase (AE; 10 min) + velocity phase (VL; 2 min), ball-specific phase (BS; 8 min) + neuromuscular phase (NM; 4 min)) would improve performance (*i.e.,* ball velocity), and reduce the perceived exertion in professional soccer players. However, no differences between WU1 and WU2 have been found, so it depends on the personal preferences of practitioners and soccer players.

## PRACTICAL APPLICATIONS

These findings could be of great interest to the strength and conditioning coaches in professional soccer teams in order to improve physical performance and perceived exertion. Specifically, ending the pre-match warm-up with a neuromuscular phase seems to be more effective to increase ball speed, ball agility and CMJ, as well as to obtain lower RPE values. However, future studies should be directed towards analyzing individualized warm-ups based on different variables, such as playing positions.

### Funding

Javier Raya-González was supported by a Ramón y Cajal postdoctoral fellowship (RYC2021-031072-I) given by the Spanish Ministry of Science and Innovation, the State Research Agency (AEI) and the European Union (NextGenerationEU/PRTR). Mehdi Ben Brahim received funding for the publication fees for this work from Prince Sultan University. The funders had no role in study design, data collection and analysis, decision to publish, or preparation of the manuscript.

### Grant Disclosures

The following grant information was disclosed by the authors:
Ramón y Cajal postdoctoral fellowship: RYC2021-031072-I.
Spanish Ministry of Science and Innovation, the State Research Agency (AEI) and the European Union (NextGenerationEU/PRTR).
Prince Sultan University.

### Competing Interests

The authors declare there are no competing interests.

### Author Contributions

- Mehdi Ben Brahim conceived and designed the experiments, performed the experiments, analyzed the data, authored or reviewed drafts of the article, and approved the final draft.
- Alejandro Sal-de-Rellán analyzed the data, prepared figures and/or tables, authored or reviewed drafts of the article, and approved the final draft.
- Adrián García-Valverde analyzed the data, authored or reviewed drafts of the article, and approved the final draft.
- Hussain Yasin conceived and designed the experiments, performed the experiments, analyzed the data, authored or reviewed drafts of the article, and approved the final draft.
- Javier Raya-González conceived and designed the experiments, analyzed the data, prepared figures and/or tables, authored or reviewed drafts of the article, and approved the final draft.

### Human Ethics

The following information was supplied relating to ethical approvals (*i.e.*, approving body and any reference numbers):

Universidad Internacional Isabel I de Castilla granted Ethical approval to carry out the study within its facilities

### Data Availability

The raw measurements are available in the Supplementary Files.

## Supplemental Information

Supplemental information for this article can be found online at http://dx.doi.org/10.7717/peerj.15803#supplemental-information.

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
