# Peer review of "The effect of three different pre-match warm-up structures on male professional soccer players' physical fitness"

_PeerJ, doi:10.7717/peerj.15803_

## Round 0.1 · original submission · Major Revisions

Thank you for submitting your manuscript to PeerJ.

I have completed my evaluation of your manuscript. The reviewers recommend reconsideration of your manuscript following major revision. I invite you to resubmit your manuscript after addressing the comments below.

When revising your manuscript, please consider all issues mentioned in the reviewers' comments carefully: please outline every change made in response to their comments and provide suitable rebuttals for any comments not addressed. Please note that your revised submission may need to be re-reviewed.

·

Basic reporting

1. The work is written correctly in English. It is understandable and easy to interpret. For me, however, there are simple mistakes at work such as: "correct article usage"; "change prepositions" ; etc. In several places, you can replace the word with a more familiar one.
In this regard, I recommend using an error correction service. (not many but they are).

2. The introduction does not exhaust the descriptions of problems related to matching an effective pre-match warm-up scheme (football). Currently, science is heading towards the area of individualization related to both matching different characteristics of the position on the pitch, and above all, taking into account individual differences.
In particular, in the described case of the senior team (age: 20.9 ± 2.4) and systematic practice: 6 ± 2 years. Therefore, the missing information should be filled in.

3. Raw data - probably not raw, it's a ready table with final results. It is described in the materials that the best sample was included in the research. so raw data should include all samples. From a scientific point of view, this gives a deep insight into the intentions of performing a given test. (This is not a complaint but a suggestion for the next work).

4. No objections to this issue

Experimental design

The material and methods of conducting the research do not allow for easy replication of the experiment in another place. There is no exact time when the tests took place in relation to the place in the training season and the day of the microcycle. We are not able to clearly read from this description that it was a warm-up before the match. At most, it was followed by a battery of tests.
It should be absolutely supplemented so that the reader can easily recreate warm-ups in similar conditions.

As for the battery of tests related to CMJ - I agree that the "MY JUMP 2.0" application is used in practice due to much lower costs compared to power platforms and that it is a validated method, however, the text of the work lacks records that the jump itself was standardized and repeatable. From the practice of everyday work in the club, I know that it takes time to master the CMJ jump pattern properly. Please complete it in the text.

The next questionable test is the "15-m ball dribbling agility test". The result of this test will always have a certain amount of chance related to the players' technical skills. With a group of only 17 players, extreme results can greatly affect the average values. If the time between the given tests was, for example, a month - the tests did not take into account the technical progress of the competitor that could have occurred. - This is missing in the description of the research methodology.

Validity of the findings

The subjective assessment of fatigue with the warm-up regimen is developed at the lower end of the scale and is between 2-4 in almost all cases (2 x 5). Drawing conclusions based on this subjective parameter is not enough. The analysis of the day's disposition related to the collected information, e.g. time and quality of sleep, muscle pain, etc., was not included.
Therefore, there may be other variables that have a significant impact on the results achieved after the warm-up, independent of the pattern of the warm-up used.

There is also no information about the power analysis of statistical tests.
-Please complete

Conclusions and practical applications should be more extensive.

Additional comments

The topics taken up by the authors of the work are indeed very important in professional sport. Finding key or uniform guidelines to apply to a larger group of football players in the world seems utopian. The warm-up will always consist of general and specialized elements, but today the requirements of the best athletes in the world are greater and require individual character and fit. (This has been included in the paragraph describing work gaps related to position on the field).

The work still requires a lot of work on the literature and perhaps supplementing the database with additional numbers that could supplement the research context.

Yours faithfully

·

Basic reporting

First of all, I would like to thank the editor for allowing me to review this study and congratulate the authors on their work. The manuscript entitled " The effect of three different pre-match warm-up structures on male professional soccer players’ physical fitness " aims to analyse the potential differences in performance resulting from three distinct warm-up structures in terms of physical performance. This topic is very interesting and within the scope of the journal and might be very useful for football practitioners. However, I have several minor concerns with this work and therefore cannot suggest its acceptance in its current form.

In general, the manuscript is well-written, with adequate language and a correct flow, making it easy for the reader to follow. I would like the authors to develop a little more the warm-up explanation. For example: What kind of stretching where included during the dynamic stretching? What does it mean for half-squat, 2 min? To maintain a half-squat in isometric position for 2 minutes? Repeated half squats for 2 minutes? I believe that a supplementary file with a complete explanation of the warm-up exercises would be very interesting for the reader.

Experimental design

No comment.

Validity of the findings

No comment.

Additional comments

INTRODUCTION
Lines 42 to 43. The reference (Krustrup & Bangsbo, 2001) is in a different font than the rest of the text.
Lines 43 to 45. Reading the sentence seems that players run several sprints over 190 m when the average football field is 100 m. Please, rewrite the sentence in order to represent the idea correctly.
Lines 46 to 47. Reference has a different font than the rest of the text.
Lines 47 to 48. Reference font.
Lines 47 to 48. The sentence does not make sense to me. Please rewrite it.
Line 51. I don’t believe is appropriate the use of “therefore” at the beginning of the sentence. I don’t think that is a consequence of the previously commented.
Lines 60. Please remove the hyphen in warm up to distinguish between the verb and the noun.
Lines 63. Reference font.
Lines 74 to 75. Reference font.
Lines 87 to 88. I believe that “the purpose of this study” or similar is missing from this sentence.

MATERIALS & METHOD
Line 111. Please add “of” between “all” and “them”.
Line 128. Please add the hyphen to warm-up.
Line 171. I believe “Statistical Analysis” should be in bold in order to follow the same style, if not, it seems it belongs to “Subjective load measure” section.

DISCUSSION
Lines 196 to 197. I am not sure if the term “main result” can be applicable to the unique two comparison that show statistical results. “Remarkable” or “notable” might be more appropriate terms in my opinion.
Line 205. AG is not defined before in the text, please do it. In the table appears like Ag rather than AG, please, be consistent with the abbreviation.
Lines 205 to 206. Authors claim that there are not statistical differences between the different warm-ups in AG. However, then they state that better results were observed for WU2. I do not understand how these two statements can go together. Moreover, I don’t really believe that a difference in the mean of 0.04 s (difference between WU2 and WU3) can be considered as better result even ignoring the statistical analysis.
Lines 207 to 211. I do not fully understand the idea that authors want to express here. Please, explain it more thoughtful in the answer to the review or rewrite the sentence.
Line 215. Reference font.
Line 226. Reference font.
Line 228. This might be a personal stay issue, but I do not consider that “After reviewing these studies, …” Is appropriate way to indicate this idea. Please, consider rewriting it.
Line 235. Founding rather than found.
Line 236. Why is that?
Line 243 to 247. I suggest authors to consider adding the biggest limitation than in my opinion this study might have. Carrying on the study in different weeks during the in-season period the physical condition of players might vary between the three different measurements. Therefore, the existence of differences or not in the results might be due to these differences rather than to the effects of the different warm-ups.
Lines 250 to 253. I do not believe that with the results of these study authors can make the statement they do in the conclusion section. It is true that WU3 might create bigger positive effects than WU1 or WU2. However there has been no analysed if the warm-up protocols create positive effects or not. In order to make that statements a 4th measurement with no warm-up intervention should have been done and showing differences with all three-warm-up protocol.

---

## Round 0.2 · accepted · Accept

My decision is to Accept your submission

·

Basic reporting

The work has been corrected as suggested. (no additional commentary)

Experimental design

The work has been corrected as suggested. (no additional commentary)

Validity of the findings

The work has been corrected as suggested. (no additional commentary)

Additional comments

The whole job looks very good now. The information contained in the work gives some indications and suggestions for motor preparation coaches on how to build a diagram and implement a warm-up protocol. Most importantly, the problem of individualization of warm-up protocols is an inspiring topic and should be very well monitored to improve the functioning of athletes not only in football. Congratulations and good luck on your next interesting research projects.